# Application of High-Pressure Homogenization for Apple Juice: An Assessment of Quality Attributes and Polyphenol Bioaccessibility

**DOI:** 10.3390/antiox12020451

**Published:** 2023-02-10

**Authors:** Krystian Marszałek, Urszula Trych, Adrianna Bojarczuk, Justyna Szczepańska, Zhe Chen, Xuan Liu, Jinfeng Bi

**Affiliations:** 1Department of Fruit and Vegetable Product Technology, Prof. Waclaw Dabrowski Institute of Agricultural and Food Biotechnology—State Research Institute, 36 Rakowiecka St., 02532 Warsaw, Poland; 2Department of Food Technology and Human Nutrition, Institute of Food Technology and Nutrition, University of Rzeszow, 2D Zelwerowicza St., 35601 Rzeszow, Poland; 3Key Laboratory of Agro-Products Processing, Ministry of Agriculture and Rural Affairs, Institute of Food Science and Technology, Chinese Academy of Agricultural Sciences (CAAS), Beijing 100193, China

**Keywords:** apple juice, high-pressure homogenization, physiochemical properties, enzyme activity, polyphenol profile, bioaccessibility

## Abstract

In the current work, the influence of high-pressure homogenization (HPH) (200, 250, and 300 MPa) on pH, Brix, turbidity, viscosity, particle size distribution (PSD), zeta potential, color, polyphenol oxidase (PPO), peroxidase (POD), polyphenol profile and bioaccessibility of total phenolic compounds was studied. The results show no change in the apple juice’s pH, TSS and density. In contrast, other physiochemical properties of apple juice treated with HPH were significantly changed. Besides total phenolic content (15% degradation) in the HPH-treated apple juice at 300 MPa, the PPO and POD activities were reduced by a maximum of 70 and 35%, respectively. Furthermore, among different digestion stages, various values corresponding to PSD and zeta potential were recorded; the total phenolic content was gradually reduced from the mouth to the intestine stage. The polyphenol bioaccessibility of HPH-treated apple juice was 17% higher compared to the untreated apple juice.

## 1. Introduction

A diet rich in fruit and vegetables containing bioactive compounds such as antioxidants, vitamins and minerals improves human health [1]. In this regard, consuming at least 400 g of fruit and vegetables per day was recommended by World Health Organization [2]. One of the ways to provide valuable nutrients from fruit and vegetables is the consumption of juices, which is becoming popular [3].

One of the world’s most commonly produced and consumed fruits are apples, with 75 million tones produced in 2018–2019 [3]. Apples are fruits that are widely available, versatile in production and affordable, in addition to being highly nutritious and containing a wide range of bioactive compounds. Based on nutritional value, some researchers consider the whole fruit more valuable. By drinking juices, the health benefits of the fruit can be effectively realized [4,5].

The high pH and sugar content of FJ forced the industry to approach suitable processing methods to prevent the growth of unfavorable microorganisms and consequently guarantee the shelf life and safety of the product [6,7]. In this context, heat treatment is common to prolong FJ’s shelf-life at an industrial scale. Unfortunately, products processed this way can cause undesirable biochemical and nutritional damage, affecting their sensory properties [8,9]. Consequently, increasing attention is being paid to the development of non-thermal methods for the stabilization of FJ. Non-thermal technological solutions are being investigated to obtain a nutritionally rich product that simultaneously has sensory properties equal to those of fresh products while guaranteeing microbiological safety. Among non-thermal food processing methods, technologies based on high pressure, such as high-pressure homogenization (HPH), are worthy of highlighting [10,11].

High-pressure homogenization consists of forcing the fluid through a valve with a very narrow, adjustable gap, which generates high pressure and a surge of flow. This process leads to a reduction in particle size and mechanical disintegration of microorganisms [12]. The HPH process also improves juice characteristics such as homogeneity and stability. With the development of technology and the upgrading of industrial homogenizers, pressure parameters and process efficiency have significantly advanced.

Applying the HPH method to FJ production aims to eliminate microorganisms, reduce enzyme activity, and improve the product’s functional characteristics [13]. Some literature reports have shown varying effects of HPH on the bioactive compound profile of processed foods [14,15,16]. Compared to the classical pasteurization method, HPH maintains more bioactive components [17,18]. Many authors have shown a positive effect of HPH on increasing the availability of bioactive compounds such as polyphenols or carotenoids [19,20,21].

Recent studies have described the effects of HPH on FJ such as blackcurrant, peach, pomegranate and mango juices, or vegetables juices, e.g., tomato, carrot [18,22,23,24,25,26]. There are several reports of HPH-treated apple juices [27,28,29], but this paper is designed to analyze and summarize current scientific knowledge on the effects of HPH on apple juice quality and polyphenol bioaccessibility. It will advance the knowledge of this modern method of non-thermal food processing and develop the rather under-researched area of the method’s impact on apple juice’s nutritional and health benefits.

## 2. Materials and Methods

### 2.1. Preparation of Apple Juice and High-Pressure Homogenization (HPH) Treatment

The apple fruit (Golden delicious var.) was washed, peeled, extracted via juice extractor (Joyong Electric Appliance Co., Jinan, China), and kept at 4 °C.

The apple juice was processed in the homogenization valve of the HPH system (JN-02HC series, Guangzhou, China) equipped with a circulating cooling system (HL-01AS, CN). The HPH treatment was carried out at a pressure of 200, 250, 300 MPa, and 2.1 L/h at room temperature. The HPH-treated apple juice was packed in 200 mL glass bottles and stored at 4 °C for further analysis [30].

### 2.2. pH, Brix, Turbidity, Viscosity, Particle Size Distribution (PSD) and Zeta Potential

Total soluble solids (TSSs), particle size distribution (PSD) and pH values were measured using a digital refractometer (WZB 45, CN), a Mastersizer 2000 (Malvern Instruments, Malvern, UK) and a portable pH meter (Testo 205, DE) at room temperature, respectively. The HPH-treated and untreated apple juices were diluted 20 times and 1 time. 

A Zetasizer Nano-ZS (Malvern Instruments, UK) and a portable Turbidimeter (Model 2100P, CN) were used to determine the zeta potential and turbidity at 25 °C, respectively [30].

### 2.3. Color

The changes in the color of HPH-treated and control were determined using a Color Quest XT colorimeter (Hunter Associates Laboratory, Washington, VA, USA) and were reported as L *, a *, and b *-values. Moreover, the total color differential (∆E) was determined using the following equation.
(1)$$∆E=({\left({L\;-\;L}_{0}\right)}^{2}+{\left({a\;-\;a}_{0}\right)}^{2}+{\left({b\;-\;b}_{0}\right)}^{2}{)}^{0.5}$$

L_0_, a_0_ and b_0_ were related to the color parameters of the untreated apple juice.

### 2.4. Polyphenol Oxidase (PPO) and Peroxidase (POD) Enzyme Activities

Terefe et al. (2010) [31], with slight modifications as follows, was used for enzyme activity determination. The extraction solution consisted of a 0.2 M sodium phosphate buffer (pH = 6.5) containing 1 M NaCl, 4% (*w/v*) polyvinylpolypyrrolidone (PVPP) and 1% (*v/v*) Triton X-100. The apple juice and extraction mixture (4.5 mL: 4.5 mL) were shaken using a vortex (IKA, Staufen, Germany) for 1 min and centrifuged (Rotina 380R, Hettich Instruments, Tuttlingen, Germany) at 11,000 × g for 30 min at 4 °C. The supernatant was centrifuged under the same conditions and used for subsequent analysis.

For the PPO assay, 300 μL of the supernatant was added to 3 mL of 0.05 M phosphate buffer (pH = 6.5) containing 0.07 M of catechol. The absorbance was measured at λ = 420 nm and 25 °C for 10 min using a UV–visible spectrophotometer (6705 UV–vis Spectrophotometer, Jenway, Eaton Socon, UK).

For the POD assay, 50 μL of the supernatant was added to 3 mL of 0.05 M phosphate buffer (pH = 6.5). The reaction was started by adding 50 μL of 1% p-phenylenediamine (*w/v*) in 0.05 M phosphate buffer (pH = 6.5) and 50 μL of 1.5% hydrogen peroxide (*v/v*). The absorbance was measured at λ = 485 nm and 25 °C for 10 min. Blank samples for PPO and POD assays were prepared using the same components, but the supernatant was replaced with a 0.05 M phosphate buffer (pH = 6.5).

The residual activity for PPO and POD was calculated according to Equation (1):
(2)
RA (%) = A/A_0_ × 100

where A is the activity of the treated juice and A_0_ is the activity of the control.

### 2.5. Polyphenol Profile

The supernatant used to determine individual polyphenols was prepared as follows: five milliliters of 80% (*v/v*) methanol containing 0.1% (*v/v*) of HCl was added to 5 mL of apple juice. The samples were treated with ultrasound for 5 min (45 kHz, 200 W, 25 °C, MKD Ultrasonic, Warsaw, Poland), then centrifuged (Rotina 380R, Hettich Instruments, Tuttlingen, Germany) at 3670 × g for 5 min at 4 °C, and supernatant was transferred to a 25-mL flask. The extraction was repeated four times and the supernatant was filtered (pore size 0.45 μm, Macherey-Nagel, Duren, Germany).

The polyphenol profile was determined using the method proposed by Tsao et al. (2003) [32], which was previously validated. Polyphenols were identified based on purchased standards from Sigma-Aldrich (St. Louis, MO, USA). A Sunfire C18, 5 μm, 4.6 mm × 250 mm analytical column with a Sunfire C18 Sentry guard cartridge, 5 μm, 4.6 mm × 20 mm (Waters) with a photodiode detector (Waters 2996, USA) was used. The column temperature was 25 °C. Samples were eluted using a gradient of 6% (*v/v*) acetic acid (solvent A) and acetonitrile–HPLC grade (solvent B), as follows: from 0 to 45 min, 100% (A); then 45–60 min, 85% (A) and 15% (B); 60–65 min, 70% (A) and 30% (B); then 65–70 min, 50% (A) and 50% (B); 70–73 min, 100% (B) and finally 73–75 min, 100% (A). The separation of the 10 μL samples was performed within 75 min at a flow rate of 1.0 mL/min. The concentration of polyphenols in the juice was calculated on the basis of the standard curve at different concentrations of standards of individual polyphenols, and the results were expressed as mg/L.

### 2.6. Bioaccessibility

According to a methodology reported by Liu, et al. (2016) [33], HPH-treated and untreated apple juice were subjected to *in vitro* digestion, including the mouth, stomach and small intestine stages. In each phase of the digestion, solution was collected to measure the PSD, zeta potential and total phenolic content (TPC). The bioaccessibility of TPC was calculated according to the equation:(3)$$\mathrm{Bioaccessibility}\;\left(\%\right)=\frac{M_{\mathrm{mc}}}{M_{\mathrm{rd}}}\;\;\;\;$$

M_mc_ and M_rd_ were the TPC in the micelle fraction (mg) and the raw digest (mg).

### 2.7. Statistical Analysis

All analyses were carried out in triplicate. The results were expressed as a mean value ± standard deviation (S.D.) and analyzed using STATISTICA 10 software (StatSoft, Tulsa, OK, USA) with one-way analysis of the variance (ANOVA). Statistically significant differences between these values were assayed using Tukey’s test at a confidence level of α = 0.05.

## 3. Results

### 3.1. Physicochemical Properties

The influence of HPH treatment on the physicochemical properties of apple juice is shown in Table 1. The increase in the pressure of HPH processing did not change pH, TSS and density. No changes were found in the pH of orange juice treated with HPH [34]. Furthermore, Yildiz Gulcin et al. (2019) reported that with increasing pressure, no change in the TSS of the HPH-treated peach juice was noted [18]. Nevertheless, the viscosity of apple juice after HPH treatment was lower at 200 and 300 MPa compared to the control. At the same time, there was an increase in the viscosity of HPH-processed apple juice at 250 MPa. The results were attributed to the fact that the pressure at 200 MPa did not affect the molecular structure. When the pressure was elevated from 200 MPa to 250 MPa, high pressure promoted the dissolution and expansion of molecules, thus resulting in a higher viscosity in the HPH-treated apple juice [35]. Nevertheless, with the pressure continuously increasing, the homogenization effect of HPH processing played a dominant role in apple juice, thereby damaging the interaction between pectin molecules [35]. In addition, the PSD of apple juice treated with HPH (Figure 1) exhibited a consistent trend with the viscosity of HPH-treated apple juice, which further confirmed the results for the viscosity of the HPH-treated apple juice. The turbidity in the apple juice processed with HPH was higher than that of the control, which was explained by the increase in suspension in the apple juice caused by the homogenization effect [36]. On the contrary, the zeta potential of HPH-treated apple juice was gradually enhanced, with the pressure increasing, which was attributed to the destruction of the original structure of pectin at a high homogenization effect. Yang Ni et al. (2019) also found that when the pressure was above 70 MPa, the cloudy ginkgo beverages’ zeta potential gradually declined as a pressure function [37].

The color changes of HPH-treated and control samples are displayed in Table 1. The L and b values of apple juice after HPH treatment at 200 and 300 MPa were lower than those of the untreated apple juice, while the a-value of the HPH-treated apple juice demonstrated a higher value. However, when the pressure was 250 MPa, the L, a and b values of apple juice treated with HPH did not significantly change compared to the control. The reduction in the L and b values of HPH-treated apple juice may be due to the decrease in PSD, resulting in the decline of light diffraction. The decrease in the L and b values of orange juice after HPH processing was also reported by Rita-María et al. (2019) [34] and Chandi et al. (2020) [35]. The increase in the a-value of HPH-treated apple juice demonstrated that the apple juice became more red after HPH processing. Furthermore, the ∆E values of apple juice treated with HPH were higher than 3, and the ∆E value first reduced and then increased as the pressure increased from 200 MPa to 300 MPa. The results indicate that HPH treatment causes higher color changes and other physicochemical properties due to increase in the pressure [31,38].

### 3.2. Enzyme Activity and Polyphenol Profile

The effect of HPH treatment on the PPO and POD activity is shown in Table 2. The PPO activity of HPH-treated apple juice at 200 MPa did not change compared to the control juice. In contrast, the residual activity of PPO in the HPH-treated apple juice gradually declined when the pressure increased from 200 MPa to 250 MPa. The highest reduction (70%) in the HPH-treated apple juice was noted at 300 MPa. Sauceda-G’alvez et al. (2021) demonstrated that the PPO activity of apple juice treated with HPH at 300 MPa was undetected [36]. Szczepańska et al. (2021) achieved 21.5% reduction of PPO in apple juice treated at 200 MPa [29]. The POD activity in the HPH-treated apple juice was reduced with increasing pressure and was lower than that of the control. Meanwhile, the lowest residual activity of POD was 65.8%. The reduction in the PPO and POD activity may be attributed to the damage to the enzyme structure caused by the high pressure [39]. Compared to the residual activity of PPO, POD showed higher residual activity in the HPH-treated apple juice, indicating that POD was more tolerant to pressure than PPO [40].

The polyphenol profile in the apple juice after HPH treatment is investigated in Table 2. The results show that the polyphenol contents in the HPH-treated apple juice gradually decreased with increased pressure and were lower than in the control juice. The phloridzin, epicatechin, chlorogenic acid, caffeic acid, gallic acid, and total phenolic compounds in the apple juice treated with HPH at 300 MPa were reduced by 63%, 48%, 12%, 16%, 22%, and 18%, respectively. The reduction may be explained by the degradation of polyphenols induced by high pressure and cavitation effect [38]. In addition, a slight decrease in total phenolic content in HPH-treated apple juice at 100 MPa was reported by Chandi et al. (2020) [30]. HPH could effectively reduce the PPO and POD activity of apple juice, which can be used as an important factor for extending the shelf-life of HPH treated apple juices. On the other hand, HPH processing causes the decline of polyphenol contents in apple juice, which significantly affects the nutritional properties.

### 3.3. Influence of HPH on Bioaccessibility of Polyphenols and Physical Parameters of Apple Juice during Simulated Digestion

The changes in PSD, zeta potential and TPC content in the apple juice under simulated digestion are shown in Table 3. In the mouth stage, the PSD of HPH-treated and untreated apple juice was larger than that of the control sample, which may be explained by the unfolding of the macromolecule chain caused by enzymes in the mouth. In the stomach stage, the PSD of HPH (200 MPa)-treated and untreated apple juice reduced compared to the mouth stage, while the PSD of apple juice treated with HPH at 250 MPa and 300 MPa enhanced. This is because high temperature promoted more components to flow out of the cell, resulting in longer digestion [41]. In the intestine stage, the PSD of both HPH-treated and untreated apple juice was reduced, demonstrating that these macromolecules were catalyzed into low-molecular-weight compounds by some enzymes in the intestine [42].

For zeta potential, the zeta potentials of HPH-treated and untreated apple juice were significantly changed when the digestion stage was from the mouth stage to the intestine stage. In the mouth stage, the zeta potential of apple juice treated with HPH at 200 MPa was enhanced, whereas the zeta potential of apple juice treated with HPH at 250 MPa and 300 MPa was reduced. When the digestion stage was in the stomach stage, the zeta potentials of both HPH-treated and untreated apple juice considerably increased. On the contrary, in the intestine stage, the zeta potentials of both HPH-treated and untreated apple juice were considerably reduced. Zeta potential has a positive correction with the pH value of apple juice [32]. The pH required for various enzymes is different at each stage of digestion [43]. Therefore, the zeta potential of HPH-treated and untreated apple juice demonstrated significant changes.

The content of total polyphenols decreased at each subsequent stage of apple juice digestion. This is due to the activity of digestive enzymes and the drastically changing pH in the simulated digestive tract. The recovery of polyphenols at the stage of digestion in intestinal conditions was slightly higher in CS (64%) and the juice after HPH at 300 MPa (60%) compared to juices after HPH at 250 MPa (53%) and 200 MPa (55%). There were significantly higher recoveries of total polyphenols at the micelle stage in HPH samples (27–32%) compared to controls (24%). The use of HPH resulted in a decrease in the average particle size of apple juices. Therefore, polyphenols could be better extracted and absorbed into the micelles to a greater extent [43]. This phenomenon may also affect the bioaccessibility of polyphenols in HPH-treated apple juice.

Bioaccessibility of total polyphenols from apple juice was determined as the ratio of their concentration in the micelle fraction to their concentration in the raw digest (Figure 2). Apple juices subjected to HPH treatment at three pressure parameters were characterized by higher (by 17%) bioaccessibility of total polyphenols compared to the control sample. Bioaccessibility of the analyzed compounds in control was 36.8%, while HPH samples reached 51.3–55.8%. The HPH treatment can damage the plant cell walls, thus increasing the extraction of phytochemicals inside the cells into the supernatant [44].

## 4. Conclusions

The physiochemical properties, enzyme activity and polyphenol profile of HPH-treated and untreated apple juice were investigated. No changes in apple juice’s pH, TSS and density were noted after an increase in the pressure of HPH processing. While the viscosity of apple juice after HPH (200 and 300 MPa treatments) was lower compared to the control, an increase in the viscosity of HPH-processed apple juice at 250 MPa was reported. In addition, a consistent trend between viscosity and PSD of apple juice treated with HPH was noted. The increase in the pressure gradually enhanced the zeta potential of HPH-treated apple juice. On the contrary, the PPO, POD activity and polyphenol profile of HPH-treated apple juice gradually reduced as the pressure increased. The L * and b * values after HPH treatment at 200 and 300 MPa were lower than the corresponding value for the CS, while the a *-value of the HPH-treated apple juice was higher. However, significant differences in L, a and b values of apple juice treated at 250 MPa were not observed when compared with CS. During the digestion period, the PSD of HPH-treated apple juice demonstrated an increase and decrease trend, while the TPC gradually reduced. The PSD during simulated digestion had a positive correlation with the pH. The polyphenol bioaccessibility of apple juice via HPH treatment increased by 17%. The obtained results can be used for the prediction of future trends in functional food production, especially for improving the bioaccessibility of nutritional compounds from fruit and vegetable products. Future works should be focused on the application of higher pressures for preservation of juices using HPH, because pressure up to 300 MPa is insufficient considering the microbial ability of fruit juices during storage time.

## Figures and Tables

**Figure 1 antioxidants-12-00451-f001:**
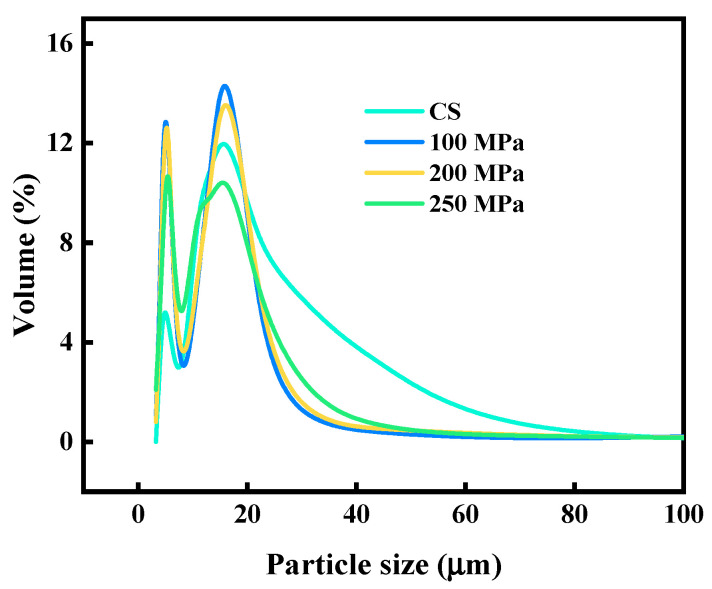
Influence of HPH processing on the particle size distribution of apple juice.

**Figure 2 antioxidants-12-00451-f002:**
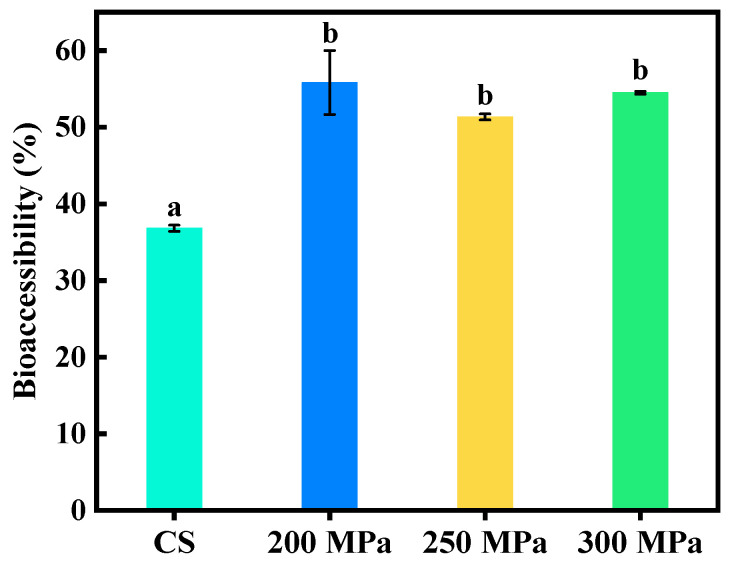
Influence of HPH processing on the polyphenol bioaccessibility of apple juice. Different letters represent the significant difference among means (*p* < 0.05).

**Table 1 antioxidants-12-00451-t001:** The physicochemical properties of apple juice treated via HPH treatment.

	CS	200 MPa	250 MPa	300 MPa
pH	3.08 ± 0.01 ^a^	3.08 ± 0.01 ^a^	3.06 ± 0.00 ^a^	3.08 ± 0.01 ^a^
^1^ TSS	13.25 ± 0.07 ^a^	13.35 ± 0.07 ^a^	13.30 ± 0.00 ^a^	13.25 ± 0.07 ^a^
Density	10.33 ± 0.00 ^a^	10.31 ± 0.01 ^a^	10.39 ± 0.00 ^c^	10.37 ± 0.00 ^b^
Viscosity (m Pa.s)	1.99 ± 0.21 ^a^	1.93 ± 0.02 ^a^	2.31 ± 0.06 ^a^	1.85 ± 0.62 ^a^
Turbidity (NTU)	93.93 ± 1.17 ^a^	133.33 ± 1.87 ^b^	119.40 ± 2.40 ^c^	178.93 ± 0.11 ^d^
^2^ PSD (μm)	38.73 ± 1.15 ^b^	24.24 ± 1.79 ^a^	28.25 ± 1.79 ^a^	22.98 ± 4.31 ^a^
Zeta potential (mv)	−18.90 ± 0.00 ^a^	−16.15 ± 0.10 ^b^	−15.50 ± 0.60 ^bc^	−14.50 ± 0.60 ^c^
L	45.42 ± 0.79 ^b^	42.27 ± 0.13 ^a^	45.21 ± 0.16 ^b^	40.30 ± 1.24 ^a^
a	7.84 ± 0.10 ^a^	8.72 ± 0.07 ^b^	7.42 ± 0.15 ^a^	9.01 ± 0.40 ^b^
b	25.67 ± 0.42 ^b^	24.97 ± 0.03 ^ab^	25.51 ± 0.21 ^b^	24.16 ± 0.79 ^a^
∆E	-	3.35	3.26	5.33

^1^ TSS: total soluble solids, ^2^ PSD: particle size distribution. Data represent the mean ± standard deviation of three replicates. Values with different letters (a–c) in the same row indicates evidently different by the pressure of HPH (*p* < 0.05).

**Table 2 antioxidants-12-00451-t002:** The enzyme activity and polyphenol contents of apple juice after being treated using HPH treatment.

	CS	200 MPa	250 MPa	300 MPa
Residual activity (%)				
PPO	100.00 ± 4.11 ^c^	100.00 ± 0.51 ^c^	49.26 ± 1.85 ^b^	29.97 ± 1.36 ^a^
POD	100.00 ± 0.98 ^c^	98.37 ± 2.90 ^c^	81.79 ± 1.10 ^b^	65.82 ± 2.44 ^a^
Polyphenol content (mg/L)				
Phloridzin	6.56 ± 0.62 ^b^	5.68 ± 0.18 ^b^	3.26 ± 0.05 ^a^	2.42 ± 0.08 ^a^
Epicatechin	17.12 ± 1.34 ^c^	13.80 ± 0.37 ^b^	12.77 ± 0.04 ^b^	8.98 ± 0.69 ^a^
Chlorogenic acid	269.57 ± 4.67 ^c^	256.67 ± 4.58 ^b^	244.34 ± 2.18 ^a^	238.23 ± 3.55 ^a^
Caffeic acid	4.44 ± 0.03 ^d^	4.24 ± 0.00 ^c^	4.07 ± 0.02 ^b^	3.75 ± 0.00 ^a^
Gallic acid	5.34 ± 0.12 ^d^	5.03 ± 0.02 ^c^	4.66 ± 0.02 ^b^	4.24 ± 0.01 ^a^
Total phenolic compounds	303.03 ± 2.80 ^d^	285.42 ± 4.79 ^c^	269.11 ± 2.21 ^b^	257.61 ± 4.15 ^a^

Data presented as the mean +/− SD (standard deviation). Different letters represent the significant difference among means (*p* < 0.05).

**Table 3 antioxidants-12-00451-t003:** The PSD, zeta potential, and TPC of apple juice treated using HPH treatment during simulated digestion.

		CS	200 MPa	250 MPa	300 MPa
^1^ PSD	Control	22.37 ± 0.00 ^ab^	19.02 ± 4.17 ^ab^	19.09 ± 1.79 ^ab^	20.42 ± 3.23 ^ab^
	Mouth	34.55 ± 6.91 ^ab^	28.51 ± 0.85 ^ab^	32.58 ± 6.89 ^ab^	36.21 ± 10.61 ^ab^
	Stomach	29.40 ± 0.00 ^b^	28.43 ± 6.55 ^ab^	44.73 ± 2.93 ^ab^	37.69 ± 2.80 ^ab^
	Intestine	23.33 ± 5.66 ^ab^	8.53 ± 2.16 ^a^	17.17 ± 0.00 ^ab^	30.72 ± 29.18 ^ab^
Zeta potential (mv)	Control	−20.45 ± 2.62 ^a^	−19.60 ± 1.72 ^abc^	−16.20 ± 3.96 ^cd^	−16.80 ± 1.27 ^bcd^
	Mouth	−20.05 ± 0.92 ^ab^	−18.35 ± 0.78 ^abcd^	−19.30 ± 0.14 ^abc^	−18.00 ± 0.14 ^abcd^
	Stomach	−9.82 ± 0.13 ^e^	−12.10 ± 1.70 ^e^	−12.55 ± 0.21 ^e^	−11.65 ± 0.64 ^e^
	Intestine	−17.30 ± 0.71 ^abcd^	−15.70 ± 0.85 ^d^	−19.30 ± 0.57 ^abc^	−18.40 ± 0.14 ^abcd^
^2^ TPC (mg/mL)	Control	1.34 ± 0.00 ^hj^	1.49 ± 0.07 ^k^	0.78 ± 0.00 ^d^	1.37 ± 0.06 ^j^
	Mouth	1.29 ± 0.00 ^ih^	1.28 ± 0.02 ^i^	1.52 ± 0.00 ^k^	1.12 ± 0.03 ^h^
	Stomach	0.68 ± 0.00 ^c^	0.97 ± 0.01 ^g^	0.84 ± 0.00 ^ef^	0.87 ± 0.00 ^f^
	Intestine	0.86 ± 0.03 ^f^	0.82 ± 0.03 ^def^	0.80 ± 0.01 ^de^	0.82 ± 0.01 ^def^
	Micelle	0.32 ± 0.01 ^a^	0.46 ± 0.02 ^b^	0.41 ± 0.01 ^b^	0.45 ± 0.01 ^b^

^1^ PSD: particle size distribution, ^2^ TPC: total phenolic content. Data presented as the mean +/− SD (standard deviation). Different letters represent the significant difference amongmeans (*p* < 0.05).

## Data Availability

The data presented in this study are available on request from the corresponding author.

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
