# Peer review of "Application of High-Pressure Homogenization for Apple Juice: An Assessment of Quality Attributes and Polyphenol Bioaccessibility"

_antioxidants, 2023, doi:10.3390/antiox12020451_

Round 1

Reviewer 1 Report

The present study focused on the influence of high-pressure homogenization on the physicochemical properties, enzyme activity, and polyphenol profile of apple juice. The topic of this study fully fills in the scope of Antioxidants. This study was designed logically and all of the results were good, but there were several suggestions for further improving the quality of the manuscript.

1.     The cultivar of the apple fruit should be provided because of the quality and polyphenol were varied in different cultivars.

2.     Section 2.5 “Polyphenol profile” should be improved. The individual standard should be provided, and the method applied for the identification of individual polyphenol should be described. Besides, the method utilized for the quantitative analysis of individual polyphenol should be validated.

3.     The HPLC chromatograms of individual polyphenol in apple juice with or without HPH treatment should be provided.

4.     Figure 2 could be removed, which could not provide important information.

5.     The format of the references should be improved.

Author Response

Reviewer #1: 

The present study focused on the influence of high-pressure homogenization on the physicochemical properties, enzyme activity, and polyphenol profile of apple juice. The topic of this study fully fills in the scope of Antioxidants. This study was designed logically and all of the results were good, but there were several suggestions for further improving the quality of the manuscript.

  1. The cultivar of the apple fruit should be provided because of the quality and polyphenol were varied in different cultivars.

Response: Thanks a lot for the reviewer’s rigorous comment. We have added the cultivar of the apple fruit to our manuscript.

  1. Section 2.5 “Polyphenol profile” should be improved. The individual standard should be provided, and the method applied for the identification of individual polyphenol should be described. Besides, the method utilized for the quantitative analysis of individual polyphenol should be validated.

Response: Thanks a lot for the reviewer’s rigorous comment. We have improved this section. In our laboratory we have most common standards from Sigma-Aldrich (St. Louis, MO, USA) in apple products e.g.: chlorogenic acid, epicatechin, catechin, p-coumaric acid, ferulic acid, gallic acid.

  1. The HPLC chromatograms of individual polyphenol in apple juice with or without HPH treatment should be provided.

Response: Thank you very much for the reviewer’s comment. The authors believe that adding a chromatogram to the manuscript does not bring anything new, especially since we have not developed new analytic method for the determination of presented nutritional compounds, therefore we do not include it in the publication. We present a chromatogram of an apple juice with the polyphenols identified in our work. If You believe that this chromatogram should be included to the manuscript we will include.

Chromatogram of polyphenols in apple juice sample. Peak identification: 1 – gallic acid, 2 – chlorogenic acid, 3 – caffeic acid, 4 – epicatechin, 5 – phloridzin.

  1. Figure 2 could be removed, which could not provide important information.

Response: Thanks a lot for the reviewer’s comments. We have deleted Figure 2 in our manuscript. Meanwhile, we have also reordered the figures of our manuscript.

  1. The format of the references should be improved.

Response: Thank you very much for the reviewer’s suggestion. We have carefully revised the format of the references.

Reviewer 2 Report

Dear DR.

Thank you for giving us the confidence to REVIEWER the manuscript in your JOURNAL. I hope to cooperate in evaluating other manuscript.

Reviewer # comments

The manuscript is good and the work and effort of the researchers is clear in it, but there are some modifications that we hope the researchers will abide by
Comments to the Author
Authors should address the following questions.

1.      Section 2.5 : line 126 : add a parentheses to (2003

2.      Section Reference :

-Line 329 : remove « pp »

              - References  32-40 : references are not regular with the previous

Author Response

Reviewer #2:

The manuscript is good and the work and effort of the researchers is clear in it, but there are some modifications that we hope the researchers will abide by Comments to the Author

Authors should address the following questions.

  1. Section 2.5 : line 126 : add a parentheses to (2003

Response: Thanks a lot for the reviewer’s comment. We have added a parenthesis in our manuscript.

  1. Section Reference :

-Line 329 : remove « pp »

              - References  32-40 : references are not regular with the previous

Response: Thank you very much for the reviewer’s suggestion. We have carefully checked and revised the format of the references in our manuscript.

Round 2

Reviewer 1 Report

The quality of the revised manuscript was improved, which could be accepted.